# Quantifying Deaths from Aspergillosis in HIV Positive People

**DOI:** 10.3390/jof8111131

**Published:** 2022-10-27

**Authors:** David W. Denning, Ellen Frances Morgan

**Affiliations:** 1Manchester Fungal Infection Group, Faculty of Biology, Medicine and Health, Manchester Academic Health Science Centre, University of Manchester, Manchester M13 9PL, UK; 2Global Action for Fungal Infections (GAFFI), 1208 Geneva, Switzerland; 3School of Medicine, University of Manchester, Manchester M13 9PL, UK

**Keywords:** *Aspergillus fumigatus*, mortality, antifungal, macrophage, neutropenia, corticosteroid, cavitation

## Abstract

*Aspergillus* spp. are ubiquitous and cause severe infections in immunocompromised patients. Less is known about its incidence and prognosis in patients with HIV/AIDS. We reviewed the mortality of invasive aspergillosis in HIV/AIDS patients. Pubmed, Embase and Medline databases were used to search for articles. Studies were excluded if they contained other aspergillosis risk factors, no original or patient survival data or were not in English. From 747 articles published, 54 studies and case reports were identified following reading, published between 1985 and 2021, with 54% papers prior to 2000 reporting 853 patients from 16 countries, none from Africa. 707 (83%) patients died with an average time from diagnosis to death of 77.5 days. Postmortem diagnosis was seen in 21% of deaths recorded. A national series from France of 242 cases of invasive aspergillosis diagnosed in life recorded a 3 month mortality of 68% pre-ART, falling to 31% after introduction of ART and voriconazole. CD4 count was recorded in 39 studies and ranged from 2 to >1000 cells/mm^3^; only 8 patients (1.8%) had a CD4 > 100 cells/mm^3^. Aspergillosis occurs in patients with HIV/AIDS and associated with high mortality but its slow progression should allow diagnosis and treatment with improved outcome.

## 1. Introduction

*Aspergillus* spp. cause several syndromes of aspergillosis of differing severity in humans [1]. Patients with advanced HIV disease (AHD) are at risk of aspergillosis, primarily invasive and the unusual obstructing bronchial forms, but also probably chronic pulmonary aspergillosis (CPA). The lungs are involved in the majority of cases of invasive aspergillosis; however the heart, sinuses, kidneys and CNS can also be affected [2]. Allergic and superficial aspergillosis (onychomycosis, fungal keratitis, external otitis) appear to be rare in HIV-infected patients although fungal keratitis with HIV infection is more frequent [3]. Invasive aspergillosis may be acute as in neutropenic or transplant recipients, but also may be subacute—which is probably more common in those with AHD. Both these entities may be lethal as may be CPA.

We published the first major series of aspergillosis in 13 HIV-infected patients in 1991 and identified a subgroup of obstructing bronchial aspergillosis [4]. Major series of aspergillosis in HIV-infected patients since 2000 were published from USA in 2000 [5], France in 2002 [6], Italy in 2002 [7], Italy in 2009 [8], Japan [9], France [10] and Russia in 2021 [11]. The literature on aspergillosis and HIV infection has been reviewed in 1994 [12,13] in 2000 [5] and again in 2016 [2], but aside from general reviews, autopsy series and case reports, the literature has been relatively silent on this topic.

The incidence of invasive aspergillosis in AIDS was estimated at 3.5 cases per 1000 person years from a study of 35,252 HIV patients in a national database [5,14]. The results of another database search of 38 million hospital diagnoses showed aspergillosis was diagnosed in 0.43% of HIV patients [2,15]. Neutropenia is a known risk factor for invasive aspergillosis, and this can occur secondary to medications used for HIV patients [14]. Neutropenia or corticosteroid use has been identified in almost half of Aspergillosis infections in patients with HIV [14,16]. Other risk factors for aspergillosis infection in patients with HIV are concomitant *Pneumocystis jirovecii* (PCP) infection and a CD4 count of less than 50–100 cells/mm^3^ [17,18,19,20].

CPA usually complicates other pulmonary disease and only in 2017 was it shown that HIV positive patients also developed CPA [21] and that the pattern of infection was similar to HIV-negative patients [22]. Many people with unproven pulmonary tuberculosis (PTB) (so called smear or GeneXpert negative TB) have CPA and not PTB [23,24], but are mis-treated and some will die as a result. From a clinical and radiological perspective, there is a substantial overlap between CPA and subacute invasive aspergillosis.

Compared with other invasive fungal infections (IFIs) less is known specifically about the incidence and prognosis of aspergillosis in patients with HIV/AIDS [8]. This is thought to be due to the difficulty in clinical diagnosis and the fact that many patients with aspergillosis remain undetected during their lifetime [8,25]. Many cases of aspergillosis are discovered post-mortem [8,25]. As such the overall impact that this fungus has on patients with HIV/AIDS is largely unknown. The aim of our work is therefore to systematically review the number of deaths in HIV/AIDS patients with aspergillosis.

## 2. Materials and Methods

The method of this review involved searching for articles published from 1985 to 2021 using Pubmed, Embase (Ovid) and Medline (Ovid) databases. Keywords surrounding three main concepts were used, these were ‘HIV/AIDS’, ‘aspergillosis’ and ‘Incidence of Deaths’. Articles including HIV/AIDS diagnosis, aspergillosis diagnosis with clinical outcome were included. Inclusion criteria involved documented cases of aspergillosis. Aspergillosis affecting any organ in the body was included. Cases were accepted as ‘aspergillosis’ if there was histopathological evidence of disease, radiological and microbiological evidence combined with other diagnoses excluded or controlled by treatment. In general, patients without an additional host factor for invasive aspergillosis as outlined by the EORTC/MSG criteria [26] were accepted as a probable case of invasive aspergillosis, if the radiological and microbiological criteria were fulfilled. No attempt was made to separate acute from subacute invasive aspergillosis or CPA. Articles including other risk factors for aspergillosis were excluded such as haematological malignancies and transplant patients, but previous or current corticosteroid use in HIV-infected patients was included as common in those with HIV. Articles were excluded if they were not original research, i.e., exclusion of other systematic reviews. Articles were also excluded if there was no confirmed diagnosis of HIV/AIDS. If an article did not include any follow up of individual cases or clear cohort data, i.e., no record of death or survival, then they were also excluded. The articles meeting the inclusion criteria were then transferred to EndNote reference manager for further analysis. Duplicates were then identified and removed. Initial screening of abstracts and titles were read and irrelevant papers removed. This was followed by a secondary screening of full texts of papers; those that were irretrievable or did not meet inclusion/exclusion criteria were removed. The results of these remaining papers formed the basis of this systematic review. 

## 3. Results

Following an extensive search of Pubmed, Embase (Ovid) and Medline (Ovid) databases using the search criteria, a total 1279 papers were retrieved (Figure 1). These comprised 339 papers from Pubmed, 646 from Embase (Ovid) and 294 from Medline (OVID). 532 papers were removed due to duplication, leaving 747 papers in total. Initial screening of titles and abstract removed 379 irrelevant papers, leaving 368 papers for full text analysis. When attempting to retrieve the full texts, 127 papers were removed due to being inaccessible (Supplementary data), although review of all these papers’ titles did not reveal any which could materially contribute to the question. Another 187 full texts were also removed as they did not meet the inclusion/exclusion criteria. This left 54 papers that were included in this systematic review (Table 1). These comprised 29 case reports, 21 retrospective studies, 2 autopsy studies, 1 clinico-pathological study and 1 medical record review (Figure 1). Most cases, if not all, were cases of invasive aspergillosis.

### 3.1. Overall Deaths

In the 54 papers, 859 patients in total were identified as having aspergillosis and HIV/AIDS. Of these 859 patients, 6 were lost to follow up, leaving 853 patients remaining to assess outcome. Of these remaining 853 patients, 707 (83%) patients died. The included studies were published between 1985 and 2021; 29 of the 54 papers (54%) prior to 2000. Only 9 out of the 54 papers (17%) of the papers were published within the last 10 years. The largest study of 228 people in the USA found 26% of patients alive at one 1 year after the diagnosis of aspergillosis, made in life. Few studies attempted to separate total mortality from attributable mortality, but co-infection and co-morbidity was common. Libanore et al. attributed 63% of their deaths directly to invasive aspergillosis [7].

The time from diagnosis to death was recorded in 33 of the 54 studies and ranged from 1 day to 18 months. The average time from diagnosis to death was 77.5 days. In this review, 13 out of 54 papers involved autopsy diagnosis of aspergillosis in HIV patients, corresponding to 147 deaths. Subsequently, 21% of the deaths reported in this study were only diagnosed post-mortem. We did not find a correlation between CD4 count and time to death after diagnosis of aspergillosis.

### 3.2. Country Distribution

Aspergillosis in HIV/AIDS has a worldwide distribution and has been previously reported in Europe, Asia, Africa and North America [13]. The studies recorded in this review occurred in 16 different countries, with the majority from the USA (n = 19). We found no cases of aspergillosis recorded in HIV/AIDS patients in Africa which fulfilled our criteria other than autopsy series, which did not fulfil our criteria (no additional data [72]). The majority of deaths (55%) were recorded in the USA (Figure 2). The least number of deaths were recorded in Switzerland, the Netherlands, Australia and Brazil (Figure 2).

### 3.3. Organ Distribution

The majority of studies in this systematic review reported pulmonary involvement. No study described outcome from CPA in HIV positive patients separately. Extension to the pericardium can cause cardiac aspergillosis or tamponade [73] with cardiac aspergillosis reported in 3 studies in this review. Renal aspergillosis can be a result of primary infection or disseminated disease [74] and was primarily reported as the primary infection manifestation in at least 4 studies in this review with dissemination to the kidneys occurring in others. *Aspergillus* sinusitis was reported in at least 3 studies in this review. The commonest site of *Aspergillus* dissemination is the central nervous system (CNS) [73]. CNS aspergillosis was reported in at least 7 studies in this review.

### 3.4. CD4 Counts

CD4 count was recorded in 39 out of 54 studies, and the results ranged from 2 to >1000 cells/mm^3^ in the 454 (53%) patients whose CD4 count was reported. Of these 454 patients, only 8 patients (1.8%) had a CD4 > 100 cells/mm^3^ (Figure 3), the vast majority had a CD4 < 50 cells/mm^3^. The study with the largest number of participants (n = 85) that recorded a CD4 count, 64% had a CD4 count of <50 cells/mm^3^.

### 3.5. Species of Aspergillus Implicated

In the case series and reports where an isolate of *Aspergillus* was obtained, 130 were *A. fumigatus* (94%) and 4 were *A. flavus*, 3 *A. niger* complex and 2 *A. terreus*. Some patients had mixed colonies. Colonization is slightly more likely to involve *A. flavus* and *A. niger* complex [56]. 

### 3.6. Diagnostic Criteria

Denis et al. found that 119 of 242 (49%) patients with HIV-associated aspergillosis did not meet the 2008 European Organization for Research and Treatment of Cancer Invasive Fungal Infections Cooperative Group (EORTC)/Mycoses Study group (EORTC/MSG) diagnostic criteria [10]. In HIV patients who did not meet the EORTC criteria, the neutrophil level was higher (2205 cells/uL), compared to the lower neutrophil count in those fulfilling EORTC/MSG criteria (705 cells/uL), however they both had low CD4 counts of 20 cells/uL and 16 cells/uL, respectively, [75]. 

## 4. Discussion

Aspergillosis is an uncommon complication of HIV/AIDS. So much so, that infection by *Aspergillus* spp. was removed from the list of AIDS-defining illnesses in 1984 [5]. It is well documented that invasive aspergillosis is more commonly associated with a reduction in neutrophils and macrophages, rather than the depletion of the CD4 T-cell population [76]. However, some studies, have indicated the incidence of aspergillosis in HIV/AIDS is clear and increasing [17,19]. The result of this systematic review supports the link between aspergillosis and HIV/AIDS patients and demonstrates a significant mortality. 

In this systematic review, 859 patients in 54 published papers were identified as having aspergillosis and HIV/AIDS and fulfilled our key criteria of documented aspergillosis and outcome information. Of the 853 patients not lost to follow up, 707 (83%) patients died. In 115 collected cases and series to 1994, 93 (81%) died, with a range from 36–100% [77]. A systematic review by Lin et al. also reported that there was a high case-fatality rate of 85.7% in patients with HIV/AIDS [78]. Similar results were seen in other studies which indicated a mortality of 70% from aspergillosis in HIV/AIDS in 19 Belgian patients (many from Africa) [79]. A national series from France of 242 validated cases of IA recorded a 3-month mortality of 68% pre-ART, coming down to 31% after introduction of ART (102). 

HIV-infected patients have defects in their innate and adaptive immunity and their macrophages show poor phagocytic function and cytokine production [80,81,82]. HIV is responsible for significantly depleting CD4 T cells and therefore reducing the population of infection-specific effector cells [80]. Dysfunction of antigen-specific cytokines in alveolar T cells in HIV patients, also provides *Aspergillus* with an advantage of avoiding respiratory immune mechanisms [80]. Finally, interferon Y produced by T lymphocytes contributes indirectly to oxidative stress and damage of *Aspergillus* hyphae [82]. So, depletion of CD4 lymphocytes in HIV increases the susceptibility to this respiratory pathogen [81,82]. 

The majority of aspergillosis cases in HIV/AIDS patients in the last 15 years have occurred in untreated patients, those noncompliant with antiretrovirals and patients with advanced immunosuppression, i.e., CD4 counts less than 50 cells/mm^3^ [2]. In a literature study of 342 cases of aspergillosis in AIDS patients, CD4 counts were reported in 140 cases; and in 133 (95%) cases, the CD4 count was <100 cells/mm^3^ [18]. Our study found a similar low (1.8%) number of patients with a CD4 count above 100 cells/mm^3^. One case study reported a HIV patient who had a high CD4 count of 863 cells/mm^3^ who survived despite having pulmonary aspergillosis [83]. It has been suggested that HIV patients, particularly those with a CD4+ count of <100 cells/mm^3^ progress faster despite current therapies [27]. This indicates a need for greater clinical suspicion of aspergillosis in HIV/AIDS patients at all stages, but particularly in those with CD4 counts <100 cells/mm^3^ to initiate prompt, sometimes life-saving treatment.

Remarkably few reports detailing CPA in HIV positive patients are published, probably because of uncertainty about the diagnosis in many instances, and the difficulty in distinguishing CPA from subacute invasive pulmonary aspergillosis. Relatively recent studies have highlighted that CPA does occur with a frequency of 6–10% during anti-tuberculous therapy [21,84], and following completion of therapy [22]. However, none of these studies provide any follow up or outcome data.

The EORTC/MSG have recently updated definitions for invasive fungal infections, but to be included as a case of invasive aspergillosis, HIV patients would have to have a proven infection or either neutropenia (which is much less common these days with dolutegravir compared with zidovudine) or prolonged corticosteroid therapy to be included as probable or possible invasive aspergillosis [26]. In this series of cases, *Aspergillus* antigen was detected in serum in 39% and in bronchoalveolar lavage in 50% of patients. Beta-D glucan was not used as a diagnostic criterion. The EORTC criteria for ‘proven’, ‘probable’ and ‘possible’ forms of invasive aspergillosis clearly have significant limitations for advanced HIV disease and often do not correlate with clinical practice. As the vast majority of cases of aspergillosis in patients with HIV/AIDS in this systematic review had a CD4 <50 cells/mm^3^, healthcare staff need to consider the possibility of aspergillosis in HIV patients with low CD4 counts (<100 cells/uL) and pulmonary lesions. The call by Denis et al. to add ‘HIV-infected individual with a CD4 count <100 cells/uL’ to the EORTC criteria to ensure that clinicians do not miss aspergillosis cases has gone unheeded [10]. There is a major need for improved diagnostic and assessment criteria for aspergillosis in HIV patients [2].

As with other forms of invasive and chronic aspergillosis, multiple approaches to diagnosis are mostly likely to yield a diagnosis. These approaches should include imaging (seeking nodules and/or cavitation in the lungs), *Aspergillus* antigen (on serum and bronchoscopy fluid, direct microscopy of sputum, BAL fluid, paranasal sinus fluid, lesion aspiration or cutaneous lesions, fungal culture (preferably high volume culture), Aspergillus IgG antibody and if possible biopsy and histopathology. The potential value of respirarory tract *Aspergillus* PCR for diagnosis remains to be explored. 

The time from diagnosis to death was recorded in 33 studies from this systematic review and ranged from 1 day to 18 months. Several studies involved in this review involved post-mortem diagnosis. The average time from diagnosis to death in the studies that recorded this data was 77.5 days, roughly 2.5 months. Very few patients survived longer than 100 days, regardless of CD4 count. A similar study reported a post-survival estimate of 2 months [7,46]. Denning et al. in one early USA study, estimated a post-diagnosis of aspergillosis survival of 4 months [4,7], while in Italy, Libanore et al. found a mean 4 week survival, and a maximum of 4 months [7]. Moreno et al. in Spain, suggested a similar but slightly longer survival time of 148 days (<5 months) in their cohort study [85] and a low 1-year survival rate of 39% [85]. A literature review study by Mylonakis et al. in the USA corroborated this, indicating 79% deaths within 6 months in 274 patients with aspergillosis and AIDS [18]. In 11 HIV patients with invasive aspergillosis in the USA, Marukutira et al. found a 30% mortality by day 30 to 42% at day 90 post diagnosis [86]. Libanore et al. attributed 63% of their deaths directly to invasive aspergillosis [7]. 

*Aspergillus* spp. is easily grown in culture from environmental specimens, but not from many clinical specimens, making it difficult to diagnose antemortem [50]. In addition to this, obtaining optimal specimens from patients is difficult as they’re often extremely unwell, and invasive procedures such as bronchoalveolar lavage may have increased risk of severe complications and death [20,85]. If *A. fumigatus* is cultured, it should be taken seriously given its dominant role in invasive aspergillosis in HIV/AIDS patients. Antinori and colleagues noticed the greatest discordance between antemortem vs. postmortem diagnosis of aspergillosis of all IFIs; only 10 of 83 cases (12%) were diagnosed before death [8]. In another study of 536 AIDS patients, aspergillosis was found at autopsy in 5.6% of cases, and only 30% of those patients were diagnosed correctly prior to death [87]. Holding et al. also found more aspergillosis cases diagnosed postmortem than antemortem, indicative of underdiagnosis and suggestive of late diagnosis in those who are diagnosed in life [5].

It is not possible from our literature review to establish whether the incidence of aspergillosis in HIV-infected patients is increasing or decreasing or static. The number of papers and cohorts published detailing aspergillosis linked to HIV/AIDS appears to be decreasing; over half the papers we identified were published prior to the year 2000 and only nine (17%) were published within the last 10 years. In a post-mortem study of fungal infections in 1630 HIV patients to 2002, invasive aspergillosis was the second most prevalent infection, which did not alter over time [8]. Denis et al. in France found a decrease over the three time periods, from 152 cases over 1992–2001 to 91 cases from 2002 to 2011 (102). 

Older reports of successful therapy of aspergillosis in HIV/AIDS are few. More recent data is more encouraging. Denis et al. demonstrated the positive impact of combined antiretroviral therapy (cART) and antifungal drugs with a 3-month survival increasing from 38% in 1992–1995 in the ‘pre-CART’ period, to 68% in 1996–2001 in the ‘cART development pre-voriconazole introduction’ period and 69% in 2002–2011 in the ‘cART with voriconazole’ period [10]. Denis et al. also noted that those diagnosed in the era after introduction of cART were more likely to be alive 3 months post diagnosis, then those HIV patients in the pre-cART era (hazard ratio [HR], 0.4 95% CI [0.3–0.7] [10]. Not only have antiretrovirals decreased mortality of aspergillosis in HIV, but Denis et al. showed that survival for the 90 HIV patients diagnosed in 2002–2011, was greater in the 59 patients who had been given voriconazole (HR for death, 0.1 95% CI [0.01–0.8] [10]. 

One significant limitation of this review was that a significant number of articles could not be accessed (Supplementary data), but no major series were omitted. Several studies were in different languages such as French, Japanese, Spanish and Russian and could not be translated and added here. Some useful data could have been missed. Many studies included in this review had limited data, e.g., one patient death in a case report. This therefore runs the risk of introducing bias. However, as the incidence of death from aspergillosis in HIV patients remains relatively low, it is expected that data sets and case reports will present only limited data, which is partly why we limited our main question to estimating mortality. Although there were 859 patients from 54 papers used in this systematic review, 228 (27%) of these patients came from a single study with conclusions mirrored by the rest of the data. Another limitation of this systematic review is that it was difficult to ascertain from the studies whether aspergillosis was the cause of death for many patients or just present at death. In some patients (as in other patient groups) aspergillosis may not have contributed to death but is more of a marker of progression of HIV/AIDS. We are unable to comment meaningfully on the radiological findings of aspergillosis in these patients, and whether they do or do not align with the EROTC/MSG precise radiological criteria. 

## 5. Conclusions

Aspergillosis is no longer included as an AIDS defining illness; however, there are numerous cases of aspergillosis in HIV/AIDS, associated with high mortality. As a result, aspergillosis should be considered in HIV/AIDS patients with fevers of unknown origin, and/or pulmonary infiltrates, especially with any cavitation. Diagnosis can be difficult and needs to be actively sought. Early diagnosis and treatment are probably important, as shown by time trends in France. As the mean time from diagnosis to death is 77.5 days, diagnosis should be possible. Only voriconazole has been subjected to any form of evaluation n this patient population and if used it substantially reduces the risk of death.

## 6. Postscript

This manuscript was written to celebrate the career of Dr David A. Stevens who was David Denning’s fellowship supervisor from 1987 to 1990 at the Santa Clara Valley Medical Centre and Stanford University. The description of 13 patients with aspergillosis complicating HIV/AIDS arose from an open, prospective study of itraconazole co-ordinated by Dr Stevens. Several patients benefitted from the first oral treatment for aspergillosis, and data from some of them contributed to the approval of itraconazole in the USA and globally in 1991. One of the patients with HIV and aspergillosis grew an itraconazole-resistant strain of *Aspergillus fumigatus* (AF90) prior to starting itraconazole therapy (which failed) and became a critically important strain for developing and then codifying susceptibility testing of *Aspergillus* spp. for later EUCAST and NCCLS (CLSI) standardised methodologies. Dr. Stevens vision in offering susceptibility testing services for fungi and immersing his group in clinical studies of new antifungal agents proved to be pivotal and global contributions to the field.

## Figures and Tables

**Figure 1 jof-08-01131-f001:**
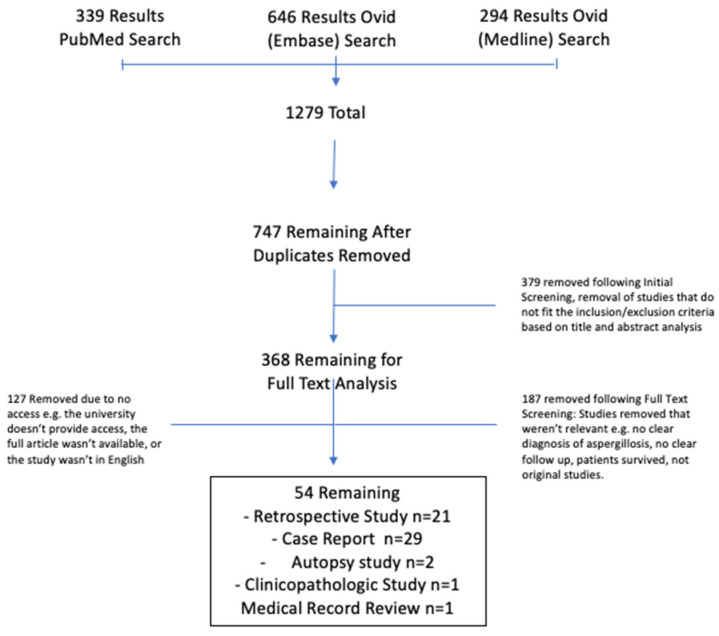
Flow chart indicated process of study selection.

**Figure 2 jof-08-01131-f002:**
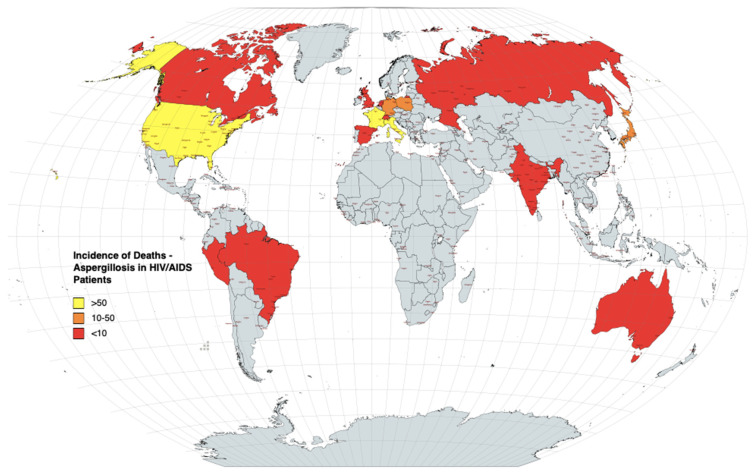
Numbers of deaths linked to aspergillosis in HIV/AIDS patients in publications by country. Categories are >50 patients, 10–50 patients and <10 patients.

**Figure 3 jof-08-01131-f003:**
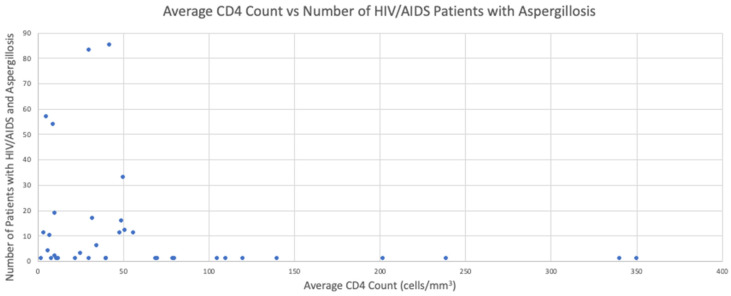
The average CD4 count vs. number of HIV/AIDS patients diagnosed with aspergillosis. Results taken from the 54 publications reviewed.

**Table 1 jof-08-01131-t001:** Summary of 54 studies of death from aspergillosis and HIV/AIDS.

No.	Reference	Year	Study Type	Country	Median CD4 (Cells/mm^3^)	Patients with HIV & Aspergillosis (n)	Patient Deaths with HIV & Aspergillosis (n)	Average Time from Diagnosis of Aspergillosis to Death
1	[27]	1997	Retrospective	USA	<100	10	4/8 (followed up)	-
2	[8]	2006	Retrospective	Italy	30	83	83	-
3	[11]	2021	Retrospective	Russia	42	85	5	-
4	[28]	1999	Case report & autopsy	-	-	1	1	-
5	[29]	1991	Case report & autopsy	-	110	1	1	31 days
6	[30]	2009	Case report & autopsy	Russia	-	1	1	-
7	[31]	2021	Case report	France	-	1	1	20 days
8	[6]	2002	Retrospective	France	5	57	31/54 (followed up)	-
9	[32]	1990	Case report and autopsy	Switzer-land	12	1	1	-
10	[4]	1999	Case reports	USA	-	13	10/12 (followed up)	3 months
11	[33]	1997	Retrospective	Canada	11	1	1	1 month
12	[34]	2006	Retrospective	Peru	-	2	2	44 days
13	[35]	1999	Case report	Spain	70	1	1	-
14	[36]	1997	Case report	USA	40	1	1	18 months
15	[37]	2009	Case report	Spain	22	1	1	78 days
16	[38]	2019	Case report	Nether-lands	-	1	1	-
17	[39]	1996	Retrospective	Germany	-	37	37	-
18	[40]	1998	Case report	Germany	40	1	1	14 days
19	[5]	2000	Retrospective	USA	-	228	228	3 months
20	[41]	2001	Retrospective	Poland	-	11	11	-
21	[42]	1994	Retrospective	UK	56.3	11	8	9.4 weeks
22	[43]	1994	Retrospective	USA	6.3	4	3	7.6 months
23	[44]	2009	Autopsy	India	-	4	4	-
24	[45]	1995	Case report	Australia	140	1	1	2.5 weeks
25	[7]	2002	Retrospective	Italy	9	54	54	1 month
26	[46]	1993	Retrospective	France	50	33	31	8 weeks
27	[47]	1985	Clinico-pathologic	USA	-	1	1	-
28	[48]	1996	Retrospective autopsy	USA	-	14	14	-
29	[49]	2009	Case report	Brazil	105	1	1	11 months
30	[16]	1994	Retrospective	USA	34.7 (n = 6)	36	24	-
31	[50]	1992	Retrospective autopsy	USA	-	18	15	-
32	[51]	1996	Case Report	Spain	69	1	1	<1 month
33	[52]	2000	Retrospective	USA	3.8	11	2	3 months
34	[53]	1998	Case reports	USA	10	2	2	11 days
35	[54]	1997	Retrospective autopsy	USA	32	17	17	-
36	[55]	2000	Retrospective autopsy	Japan	51	12	12	-
37	[56]	1992	Medical record review	USA	49 (n = 16)	45	41	7 months
38	[57]	1995	Case report	Canada	-	1	1	1 month
39	[58]	2011	Case report	-	239	1	1	1 month
40	[59]	2008	Case report	USA	30	1	1	6 weeks
41	[60]	2017	Case report	UK	>1000	1	1	1 day
42	[61]	2005	Case report	-	340	1	1	2 weeks
43	[62]	2019	Case report	-	120	1	1	3 months
44	[63]	2006	Case report	India	202	1	1	-
45	[64]	1995	Case reports	USA	25	3	3	-
46	[65]	2017	Case report	USA	350	1	1	3 days
47	[66]	1993	Retrospective	Italy	48	11	9	1.3 months
48	[67]	2005	Case report	-	80	1	1	3 months
49	[68]	2005	Case report	Brazil	8	1	1	3 months
50	[20]	1998	Case-control	USA	10	19	19	30 days
51	[19]	1998	Retrospective	Germany	7	10	9	19.5 days
52	[69]	1995	Autopsy	USA	-	2	2	-
53	[70]	2005	Case report/ autopsy	-	79	1	1	15 days
54	[71]	1996	Case report	USA	2	1	1	<1 month
						Total = 859	Total = 707	

## Data Availability

See references and supplemental data.

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
