# Peer review of "Quantifying Deaths from Aspergillosis in HIV Positive People"

_jof, 2022, doi:10.3390/jof8111131_

Round 1
Reviewer 1 Report
It's interesting to note that Aspergillosis was excluded from the list of AIDS-defining infections as early as 1984, I presume the authors are not suggesting a reinstatement? They urge consideration of the condition and I feel this is sufficient. The improvements in diagnosis and management of Aspergillosis are acknowledged and I was surprised these could not be more firmly associated with a lower mortality rate in recent years. The methodology of the analysis is sound; 1,279 papers with only 54 included and a final number of 853 patients is impressive in terms of validity. Neutropenia is listed as a significant predisposing factor for Aspergillosis; this is a much less common side effect of modern day ART than than in the earlier years: eg Dolutegravir versus AZT.
Author Response
Thank you. We have added the comment about the less frequent incidence of neutropenia
Reviewer 2 Report
This is a systematic review of the deaths from aspergillosis in people living with HIV-AIDS. This is an important topic with a well-defined and relevant aim. However, it is noticeable that the methodology presents some inconsistencies in its organization, which does not seem to make the manuscript mature enough to be published in this version.
The description of the methodology should be clearer and more complete . . I suggest better detailing the step-by-step in which the study was conducted and clarifying the inclusion and exclusion criteria.
First, all inclusion criteria must be presented and then, based on the studies that would initially be included, the exclusion criteria are presented. For example: If only studies published in English are included in this review, studies published in other languages will not be excluded as they did not meet one of the inclusion criteria.
Lines 69-72: I wonder if only manuscripts written in English are included in this study. If so, it would be interesting if you included such information as an inclusion criterion.
Lines 88: How many evaluators were involved in this stage of the study? It is important that more than one evaluator carry out this data collection to minimize bias.
Figure 1: I recommend that the flowchart be improved since this is confusing at some points.
A significant number of articles were excluded from the study. How many of them were unavailable and how many were not written in English? I suggest including only studies written in English as an inclusion criterion and articles not available in open access as an exclusion criterion.
Another suggestion: you could review the reference lists of review articles (excluded from this study) in order to rescue possible articles not found in the initial search and enrich this systematic review.
Lines 88-89: Were irrelevant articles or articles that did not meet the inclusion criteria removed?
Table 1: Wouldn't the reference number 42 mentioned as AUTOPSY be a description of a series of cases (case reports)? Same for reference number 67.
Figure 3: Do the articles mentioned describe the characteristics that justified the deaths of patients without severe immunosuppression (CD4 count of >200 cells/mm3)? If there is this information, I suggest inserting it in the manuscript.
Lines 205: As the studies did not address the association of aspergillosis with tuberculosis, a fact well reported in non-HIV individuals, I suggest bringing up this discussion.
Lines 240-243: The references mentioned are outdated and today we have more modern methods, such as galactomannan antigen detection. It would be important to bring this limitation and the advances in the rapid diagnosis of aspergillosis.
Lines 251-252: Could this information not be related to advances in antiretroviral therapy?
Lines 274-276: In the literature, is there any study comparing mortality from aspergillosis in people living with HIV-AIDS and non-HIV?
Author Response
Thank you for the suggestion. We have gone through the case reports and series and totted up the different species implicated, omitting those series where multiple different IA patient groups were included..
We have added an additional paragraph to the results.
We have also added some additional commentary on how to establish the diagnosis in the discussion section. This cannot be considered definitive given the nature of the data we found published.
Reviewer 3 Report
The manuscript addresses a relevant subject that is well structured and provides a timely review on a neglected fungal infection in HIV-positive patients. My only criticism is related to the lack of mycological data. I know it is a literature revision and it is possible the information is not available in all cases, but when possible, it would be useful to include the fungal species causing the disease, and the antifungal scheme to treat the infections. The inclusion of serological, molecular, or mycological criteria to establish diagnosis will level up the already high quality of this work.
Author Response
We have added information on species implicated (strong predominance of A. fumigatus). We have added emphasis to this in the discussion, where we discuss culture results. This fumigatus predominance should support the use of non-culture based diagnostics.
Round 2
Reviewer 2 Report
This is a systematic review of the deaths from aspergillosis in people living with HIV-AIDS. This is an important topic with a well-defined and relevant aim. However, it is noticeable that the methodology presents some inconsistencies in its organization, which does not seem to make the manuscript mature enough to be published in this version.
The description of the methodology should be clearer and more complete . . I suggest better detailing the step-by-step in which the study was conducted and clarifying the inclusion and exclusion criteria.
First, all inclusion criteria must be presented and then, based on the studies that would initially be included, the exclusion criteria are presented. For example: If only studies published in English are included in this review, studies published in other languages will not be excluded as they did not meet one of the inclusion criteria.
Lines 69-72: I wonder if only manuscripts written in English are included in this study. If so, it would be interesting if you included such information as an inclusion criterion.
Lines 88: How many evaluators were involved in this stage of the study? It is important that more than one evaluator carry out this data collection to minimize bias.
Figure 1: I recommend that the flowchart be improved since this is confusing at some points.
A significant number of articles were excluded from the study. How many of them were unavailable and how many were not written in English? I suggest including only studies written in English as an inclusion criterion and articles not available in open access as an exclusion criterion.
Another suggestion: you could review the reference lists of review articles (excluded from this study) in order to rescue possible articles not found in the initial search and enrich this systematic review.
Lines 88-89: Were irrelevant articles or articles that did not meet the inclusion criteria removed?
Table 1: Wouldn't the reference number 42 mentioned as AUTOPSY be a description of a series of cases (case reports)? Same for reference number 67.
Figure 3: Do the articles mentioned describe the characteristics that justified the deaths of patients without severe immunosuppression (CD4 count of >200 cells/mm3)? If there is this information, I suggest inserting it in the manuscript.
Lines 205: As the studies did not address the association of aspergillosis with tuberculosis, a fact well reported in non-HIV individuals, I suggest bringing up this discussion.
Lines 240-243: The references mentioned are outdated and today we have more modern methods, such as galactomannan antigen detection. It would be important to bring this limitation and the advances in the rapid diagnosis of aspergillosis.
Lines 251-252: Could this information not be related to advances in antiretroviral therapy?
Lines 274-276: In the literature, is there any study comparing mortality from aspergillosis in people living with HIV-AIDS and non-HIV?